# The CBCT Retrospective Study on Underwood Septa and Their Related Factors in Maxillary Sinuses—A Proposal of Classification

**DOI:** 10.3390/jpm13081258

**Published:** 2023-08-14

**Authors:** Kamil Nelke, Dorota Diakowska, Monika Morawska-Kochman, Maciej Janeczek, Edyta Pasicka, Marceli Łukaszewski, Krzysztof Żak, Jan Nienartowicz, Maciej Dobrzyński

**Affiliations:** 1Maxillo-Facial Surgery Ward, EMC Hospital, Pilczycka 144, 54-144 Wrocław, Poland; 2Academy of Applied Sciences, Health Department, Academy of Silesius in Wałbrzych, Zamkowa 4, 58-300 Wałbrzych, Poland; kzak@ans.edu.pl; 3Department of Basic Sciences, Wroclaw Medical University, Chalubinskiego 3, 50-368 Wroclaw, Poland; dorota.diakowska@umw.edu.pl; 4Department of Head and Neck Surgery, Otolaryngology Medical University in Wrocław, Borowska 213, 50-556 Wrocław, Poland; mkochman@mp.pl; 5Department of Biostructure and Animal Physiology, Wrocław University of Environmental and Life Sciences, Kożuchowska 1, 51-631 Wrocław, Poland; maciej.janeczek@upwr.edu.pl; 6Department of Anaesthesiology and Intensive Care, Sokołowski Hospital, Sokołowskiego 4, 58-309 Wałbrzych, Poland; marceliluk@gmail.com; 7Private Practise of Maxillo-Facial Surgery, Romualda Mielczarskiego 1, 51-663 Wrocław, Poland; nienartowicz@gmail.com; 8Department of Pediatric Dentistry and Preclinical Dentistry, Wrocław Medical University, Krakowska 26, 50-425 Wrocław, Poland; maciej.dobrzynski@umw.edu.pl

**Keywords:** maxillary sinus, Underwood septa, sinus floor, sinus lift, maxillary hypoplasia

## Abstract

**Introduction.** The presence of bone septum in the maxillary sinus is one of the most common anatomical findings. So-called Underwood septa (US) are an atypical bone formation in the maxillary sinuses. Mostly they are quite easily found in CBCT studies and have major importance in sinus lift procedures in dental surgery. Furthermore, the shape, location, and size of the bony septa are important in each maxillary sinus surgery. **Material and methods.** A retrospective study of 120CBCT scans from the authors’ own database was conducted. **Results.** Approximately 37.5% of each CBCT was associated with the occurrence of US, while just 25% had a full septum, and a total of only 14 patients had a half septa. More females have US, while healthy pneumatized maxillary sinus is most commonly found (82.22%). There is no correlation between the occurrence of silent sinus syndrome (*p* = 0.174), mucosal thickening (*p* = 0.325), or retention cyst formation (*p* = 0.272). Most sinuses are without any opacification in CBCT evaluation (91.11%), while other syndromes are not statistically relevant. **Conclusions.** It seems that the occurrence of Underwood septa is not statistically related to any clinical, radiological, or pathological condition within the sinus (*p* > 0.05). Furthermore, a more full or partial appearance of US was found in female patients.

## 1. Introduction

The Underwood septa (US) was first described by Underwood in his anatomical study on maxillary sinuses (MS) [1]. The US is a part of a thin bone wall-shaped structure inside the maxillary sinus, with various differences in shape, size, position, and bone volume. Its prevalence varies from 12–32% or more, depending on the studies [1,2,3,4,5,6,7,8]. The differences in MS anatomy might influence the existence of mucous retention cysts, mucosal thickening, or other pathologies [2,3]. This situation might happen when a complete septum can divide the entire sinus into two or more separate compartments, where the mucosal ciliary flow might be disturbed, and secondly, such anatomical disturbances might cause technical problems in sinus lift dental procedures [4,5,6]. The understanding of sinus anatomy is essential for each surgeon and clinician to estimate and properly prepare the scope of treatment. Nowadays, classic radiographs are less common in the evaluation of maxillary sinus anatomy. The condition of each maxillary sinus can be clearly and precisely evaluated in any CBCT (cone beam computed tomography studies) scans [2,3,4,7,8].

Some studies tried to evaluate the location, presence, and scope of maxillary sinus septa. Old studies distinguished those bone septal crests as those located in the anterior (between the premolar/first molar), middle (between the first/second maxillary molar), and posterior part (distal from the third upper molar) of the sinuses [1,9]. Kocak et al.’s report on 2000 CBCT studies enrolled a total of 500 cases of maxillary sinuses which had been evaluated in axial sections and described the following septal variations: the anterior septa, middle septa, and posterior septa, divided into the septal morphology as completed septa or partial septa, and their orientation, namely the transverse septa, sagittal septa, and atypical septa. On the other hand, Lee et al.’s study indicated that there can be one, two, or even more septa, that can divide each sinus into compartments either in one maxillary sinus or both [9,10]. On the other hand, Krennmair et al.’s classification distinguishes the primary septa (during maxillary development), and secondary (related to maxillary floor pneumatization and the process of tooth loss) [11]. Some other maxillary septa classifications and variations can be described according to their thickness, angulation, shape, or other components [5,6,7,8,9,10,11,12,13,14,15]. The current proposal on maxillary septa indicates that the sinus septa can be divided into primary septa and secondary septa; the primary septa arise from the development of the maxilla, whereas the secondary septa are said to arise from the irregular pneumatization of the sinus floor following tooth loss [13,14,15,16]. In other words, primary septa are congenital, and secondary septa are acquired. The septa above the apical area of an edentulous ridge cannot be distinguished into primary or secondary septa without previous radiographic records. Therefore, it can be said that septa above teeth are primary, while the septa above an edentulous ridge are primary or secondary. Because of the following, it is worth noticing that each maxillary sinus wall can have also anatomical variations, shapes, and volume, which could be related to maxillary sinus septa and their relation towards the floor of the sinus, bony walls, their position regarding the dental roots and scope of sinus pneumatization, or even others [5,6,7,8,9,10,11,12,13,14,15,16,17,18,19].

Underwood septa are most often responsible for any unsuccessful maxillary sinus lift procedures. The main issue is related to possible mucous membrane perforation in the sinus floor (The Schneiderian membrane). The importance of their identification in CBCT is very important, especially if radiological evaluation is related to future surgeries. In some cases, the scope of bone septa and their proximity to dental roots can mimic a cyst when they are not closely evaluated before any surgery. Any surgery related to those bony septa should be made with caution to protect the infraorbital artery, the posterior lateral nasal artery, and the posterior superior alveolar artery (PSAA) which grants the blood supply to the maxillary sinus [12,13,14,15,16,17,18,19,20].

The usage of CBCT greatly improved the diagnosis of each MS pathology and grants each surgeon more insights into the maxillary antrum. The presence of teeth and complete full dentition or partial/complete edentulism can also impact not only the maxillary bone shape and size but also MS findings in CBCT. Salari et al.’s study on 140 CBCT scans evaluated the sinuses for the presence of the most common pathologies, such as the occurrence of healthy pneumatized sinus and the mucosal thickening of >5 mm, retention cyst, partial or complete opacification of the sinus, and polypoidal mucosal thickening [21]. It seems that also other pathologies within the MS, such as the presence of dental septa, bone septa (Underwood septa), osteomeatal complex patency (OMC), mucocele formation, presence of cyst/dentigerous cysts, CRS (chronic rhinosinusitis), presence of SSS (silent sinus syndrome—the imploding antrum, asymptomatic chronic sinusitis with total or partial sinus opacification and sinus walls asymmetry) or others can be found [17,18,19,20,21,22].

In the presented retrospective radiological CBCT study, the authors tried to investigate the patterns of maxillary sinus bone septa and present their own proposal for this anatomical variation classification based on CBCT axial scans, along with a correlation of co-existing maxillary sinus features with the US.

## 2. Material and Methods

### 2.1. Study Design

The following research is a retrospective study on CBCT radiographs from the authors’ own clinical practice database between 2020–2023. Maxillary sinuses were studied in adults >18 years of age. These patients had their CBCT performed because of various reasons, such as orthognathic surgery, condylar hyperplasia, mandibular cyst/tumors, wisdom teeth removal, evaluation of bone/mandible asymmetry, orthodontics, sinus lift procedures, implant planning, guided bone-regeneration planning, and other procedures. Unnecessary exposure for CBCT analysis was not performed. Inclusion criteria for the study are as follows: CBCT results only from the authors’ database; patients treated, diagnosed, or operated on with CBCT 20 × 20 full scope of images; no relevant medical history concerning the maxillary sinus surgical treatment.

The exclusion criteria are as follows: CBCT from outside of the authors’ database; missing radiological/clinical data; CBCT after any possible surgery (such as Lefort I, Caldwell–Luc procedure, cystectomy, trauma/fracture cases, or others); patients not related with the scope of maxillary sinus studies.

The research respected the ethical principles of the Helsinki Declaration (2008) and the CBCT guidelines. For this research to begin, ethical approval was firstly signed by the Bioethics Committee nr 4-BNR-2022

### 2.2. Cone Beam Computed Tomography Characteristics

All of them were CBCT 20 × 20 FOV (field of view) imaging protocols based on RayScan S 5471.3 mGy (RayCompany Co., Ltd., Samsung 1-ro, Hwaseong-si, Gyeonggi-do, Repulic of Korea). CBCT evaluation was evaluated in RAYSCAN S with FOV 20 × 20 cm without slicing, with an average thickness between 0.070 mm to 0.3 mm. Exposure time was 16 s with average voltage of 4–17 mA and current of 60–90 kV. All CBCTs had been made with the same technique and equipment. A healthy sinus appearance is visible in Figure 1.

The authors described each maxillary sinus septal and bone disturbance only in axial planes of the CBCT to ensure that the presented authors’ proposal will be a good, simplified, and fast diagnostic method for this bone anatomical variation. The authors focused only on the presence of each septum according to their shape and location in either vertical, horizontal, or combined appearance in CBCT axial scans. The reference point was situated from the bottommost point of each maxillary sinus towards one centimeter superiorly in vertical sinus diameter in each CBCT. Bone angulation, bone thickness, shape, relation with dental tooth apexes, position between teeth apexes/maxillary sinus walls, or similar factors, such as septal height, do not have an influence on the proposed classification.

### 2.3. Methods—Classification Proposal

The authors’ proposal on maxillary sinuses with Underwood bone septa classification was based on 120 CBCT 20 × 20 axial scans (Figure 2). The following classification proposal is shown: (A) maxillary sinus without any bone/septal disturbances; (B) one septum in the anterior part of the sinus; (C) one septum in the posterior part of the sinus (Figure 3); (D) a single septum in the middle of the sinus (Figure 4); (E) two septa dividing the sinus into three components (Figure 5); (F) a vertical septum that divides the sinus into right and left, equal or not, parts (Figure 6); (G) located in part of the sinus, either anterior/posterior/middle part with some septa mimicking a clover leaf appearance (Figure 7); (H) maxillary sinus divided with multiple septa in a chaotic matter, like the spider-web appearance; (I) a single not full septum but a half septa in the maxillary sinus; (J) two half septa; (K) three half septa; (L) four or more half septa placed randomly in the maxillary sinuses (Figure 2, Figure 8 and Figure 9).

### 2.4. Statistical Analysis

Statistical analysis was performed using Statistica v. 13.3 (Tibco Software Inc., Palo Alto, CA, USA). Descriptive data were presented as a number of observations (percent) or mean + standard deviation (+SD). The chi-square test, Fisher’s exact test, and Student’s *t*-test were used for the analysis of the relationship between Underwood septa status and demographical, clinical, or radiological features. A two-tailed *p*-value of < 0.05 was considered statistically significant.

## 3. Results

### 3.1. Relationship between Demographical, Clinical, or CT/CBCT Radiological Features and Underwood Septa Status

Of the 120 CBCT radiographs evaluated, a total of 37.5% were only related to both full (*n* = 31) and/or partial bony septa (*n* = 14) in the maxillary sinus cavity (*p* > 0.05) (Table 1). Most MS were fully pneumatized and healthy, while no gender- or age-related correlation was noticed (*p* > 0.05). The evaluated patient’s CBCT database dental status was not significant and mostly related to a healthy sinus (*p* = 0.117). The SSS remains a rare entity not related to the sinus septa (*p* > 0.05). The partial MS opacification level was not correspondent with either mucous retention cysts or mucosal thickening but was possibly associated with polypoidal mucosal thickening in the patients. The following suggests that the number of teeth present in the dental arch and the patient’s age do not correlate with the septal occurrence along with additional symptoms (Table 1 and Table 2) (Figure 10). The occurrence of CRS and degree of mucosal thickening were not related to the presence of full or partially developed bone septa within the sinus (*p* > 0.05). All gathered results suggest that there is no statistical correlation between both full and/or partial septal walls within the maxillary sinuses (*p* > 0.05). Furthermore, there is no relation with other evaluated symptoms, such as the volume of mucosal thickening, polypoidal formation, retention cyst occurrence, SSS presence, nor the volume of sinus opacification and chronic rhinosinusitis (*p* > 0.05). It seems that the occurrence of a healthy sinus without any septal walls (*p* < 0.05) and the occurrence of a single full septum in the anterior part of the MS are the most common findings (*p* < 0.05). The following results indicate that the presence of either full or partial US in MS is very random and individually related to each patient’s sinus anatomy. Each clinician should be aware of various differences in the maxillary sinus anatomy.

### 3.2. Classification Proposal

The occurrence of Underwood septa is mostly related to its accidental findings in CBCT. A normal Underwood septa is classified as a full septa, arising between two or more walls of the maxillary sinus. Because their presence, shape, and location are random, many authors have tried to study them and many possible classifications systems are known. During the authors’ retrospective study on CBCT, not only fully shaped septa were studied, but also the presence of half or not fully evolved septal walls inside the maxillary sinus. The occurrence of both bone formations was studied according to the location and co-existing radiological, clinical, and surgical factors. Authors tried to study the mentioned septa in the CBCT axial scans only to determine and establish an easy classification proposal. The authors’ suggestion on those anatomical variations were later correlated with co-existing factors to establish any possible connection or clinical importance and to statistically measure other possible co-existing factors. As presented in Figure 2, the authors’ retrospective study on maxillary sinuses contributed to the creation of the authors’ own classification proposal on US in MS.

### 3.3. Clinical and Surgical Cosiderations

The presence of full or partial Underwood bone septa in maxillary sinuses might influence the result of some surgeries within the sinus itself. A half septa occurrence might lead to a bigger possibility of a torn in the lifted mucous membrane of the sinus, while a full septa can be a significant anatomical obstacle to perform a good lift procedure (Figure 10). While considering the sinus lift procedure and dental implant placement, the occurrence of septa might cause Schneiderian membrane perforation and lead to sinusitis, while the bone material might be pushed to the sinus. This problem can be easily avoided when the surgical access to the lift is performed by not only a classical vestibular (buccal) access but also by palatal access as well. Secondly, the placement of the dental implant might also be changed to avoid its malposition under or between the septa, leading to a very complicated sinus lift procedure. Furthermore, the shape and size of the implant can be directly related to its positioning in the sinus and the relation with the US. Thirdly, mucous retention cysts might tend to be present in some septal recesses. This might be related to decreased mucous drainage and the presence of anatomical disturbances as the septal vertical growth pattern can be found. The FESS/ESS (functional endoscopic sinus surgery or endoscopic sinus surgery) or Caldwell–Luc procedures might have some limitations if a foreign body, like the remnants from root canal fillers, are pushed inside the sinuses. Such septa might cause limited access and visibility to such foreign material. Lastly, when performing a Lefort I osteotomy, the presence of such bone septa might require some additional cutting to avoid any uncontrolled fracture lines. In cases of pharmacological treatment in sinusitis, such anatomical disturbances in sinus outflow might reduce the sinus ventilation and treatment outcomes.

## 4. Discussion

The radiological findings of each CBCT evaluation are quite important for each clinician. The presence of Underwood septa might be one of the diagnosed situations in the maxillary sinuses, followed by SSS, accumulation of mucus, formation of a retention cyst, swelling of the mucosa of the maxillary sinus floor, lack of good oroantral drainage, changes in the OMC (osteomeatal complex), or poor sinus ventilation. Each of these findings might impact any further dental procedures, sinus surgery, or other surgical approaches planned in the maxillary sinuses [12,13,14,15,16,17,18,19,20,21,22,23,24,25,26,27,28]. According to Shahidi et al., the amount of maxillary sinus variations in CBCT are common findings, but only their careful radiological evaluation might influence later good clinical and surgical outcome [12]. Most authors, like Pommer et al. or Ulm et al., focuses on the sinus lift procedure as one of the most important topics related with the US in MS [2,8]. The authors fully agree that detailed MS evaluation grants a more favorable outcome in any surgical procedures in their area. On the other hand, further studies on the correlation between MS lift procedure, from both the buccal or palatal approaches, should be discusses and compared with US localization in future papers.

The Underwood septa (US) can be present in different shapes, sizes, lengths, thicknesses, and locations, while a single septum is the most common one, followed by two septa and rarely more [3,4,5,6,7,11,12,13,14,15,16,17,18,19,20,21,22,23,24,25,26,27,28,29]. Their occurrence was associated with many possible etiological factors but which were still random in shape and size, although a possible correlation with dental roots and the height of maxillary alveolar bone process is also worth noticing [15,16,17,18,19,20]. Each septum or half septa can be located more sagittally, transversally, horizontally, and in mixed forms, or be atypically placed in the maxillary sinus. Each septal fragment in the maxillary sinus can be either completed or partial, connected to either the inferior, medial, or lateral wall of the sinus, or be a combination of all, which is a less common anatomical variation [22,23,24,25,26,27,28]. When presented with bony septa dividing each maxillary sinus into anatomical compartments, in some situations, a possible correlation with possible retention cyst formation might be found [8,9,10,11,12,13,14,15,25,26,27,28,29,30]. Many studies tried to describe the patterns of each bone septa, but it appears they are randomly placed, and as of yet no adequate studies exist on this matter. Similarly, a systematic review on the US by Malec et al. indicates that the US can be found in 9% to 70% of patients (with a mean prevalence of about 36%) which is a very wide range of occurrence, as with their shape, size, and location [18,19,20,21,22,23,24,25]. On the other hand, the presence of retention cysts (or mucous retention cysts, MRCs) in the maxillary sinuses (RCMs) was reported to be associated with the shape, deviation, and position of the nasal septum according to some studies; however, their formation regardless of the presence of US is not yet clear [20,21,22,23,24]. Most cysts, on the other hand, are found accidentally (22–36%) on the diagnostic MRI of head/brain scans [20,21,22,23,24,25,26,27,28,29,30,31,32,33,34,35,36,37]. Similar findings are confirmed by the presented paper. Mostly US are presented in a healthy MS or are associated with mucous retention cysts or some degree of mucosal thickening, which corresponds with studies by Salari et al., Raghar et al., and Wang [21,33,37]. A special atypical findings includes the occurrence of a silent sinus syndrome, which still is a rare finding, especially when co-existing with some US, as confirmed by Manila et al. and presented in the study by the authors [22].

So far, few studies have tried to evaluate data retrospectively, to shape the position and occurrence of the US, and to establish a proposal for their classification. In order to fully establish some classification, most authors set some anatomical landmarks, such as each of the maxillary sinus walls, the anterior and posterior nasal spine, the nasal septum, or others [24,25,26,27,28]. Naitoh et al. found that some of the septa in the anterior maxillary sinus region were anterolaterally directed and positioned, while some from the interior wall and most of the septa in the transverse palatine suture region were laterally directed from the interior wall. The measurements between the anterior and posterior nasal spine (ANS-PNS) are quite useful [25,27]. Some other authors, like Munetaka et al., measured the angulation of the septa depending on some anatomical reference points [28]. From the authors’ point of view, the shape, size, angulation, and position of a full or partial US is random. Some studies, like Malec et al., Naitoh et al., and Hong et al., seem to confirm the authors’ hypothesis on random patterns of US; however, some further studies on teeth eruption, sinus hypo/hyperventilation cases, and osteomeatal complex patency needs to be scheduled in order to exclude or confirm their role in the occurrence of Underwood septa [24,25,26].

The presence of the US can be troublesome, especially in dental implant placement procedures with a sinus lift approach. Authors, such as Alhumaidan et al. in their study of 178 CBCT scans, conclude that dental implant placement combined with sinus lift procedures with Schneiderian membrane lift has higher possible failure rates than a sinus lift procedure on a healthy MS [29]. In cases of severe bone atrophies, the placement of any dental implant is related with adequate sinus lift procedures and the placement of patient-specific dental case implants dedicated to each anatomical situation in the atrophic bone, which is confirmed by studies by Costa et al. and Manji [27,34]. The authors’ study was based on the study using a modified Al-Faraje’s classification into VII septal patterns [30]. On the other hand, Sigaroudi et al.’s study on 222 CBCT images found that the shape and position of septa in the molar region were more common than in the premolar area, which is also related to the increased difficulty of any surgical procedure in the maxillary sinus [25,26,27,28,29,30,31,32,33,34,35,36,37,38]. Since dental implant procedures are quite common, they should be carefully planned to avoid unnecessary complications, like sinusitis, oroantral fistula, or others. From the authors’ perspectives, the knowledge on sinus anatomy and radiology in CBCT studies is essential.

The presence of atypical maxillary sinus anatomy and the occurrence of bone septa greatly influence each dental surgery implantology case. The US not only influences dental implant placement but might also be a source of potential mucous accumulation, and because of their shape, size, and proximity to dental roots and bony walls, US can mimic a cyst or a dentigerous cyst. This situation is mostly associated with the septal position, their curve, thickness, and proximity to either a wall or a tooth apex. The US sinus septa characteristics were studied by Assari et al. in a CBCT study based on three planes: sagittal, coronal, and axial [5,30,31,32,33,34,35,36,37,38,39]. Despite the presence of US, the accumulation of mucous, mucosal thickening, and retention cysts were also found. Secondary maxillary sinus findings might include a decreased oroantral complex drainage, total sinus opacification, and polypoidal mucosal thickening. Therefore, any abnormalities or atypical anatomy in the maxillary sinuses should be carefully studied in CBCT scans [35].

Limitations of the study include the small number of CBCT scans in the database and patients with various scopes of diseases and syndromes within the facial skeleton diagnosed in CBCT; the corresponding number of patients with full/partial dental arches or totally edentulous jaws for comparison purposes; the necessity of improving the study groups to those scheduled for a sinus lift procedures, and those who required a otolaryngological treatment to compare the findings, and lastly if the amount of data was adequate on how retained/partially erupted maxillary teeth influence the sinus volume and septa/half septa occurrence in young adults.

## 5. Conclusions

Maxillary sinus variations in shape, size, and presence of bone septa can be easily diagnosed and studied in each CBCT evaluation. The occurrence of full or partial septa in the sinuses can be classified in many possible ways. Some of them can be associated with some clinical and radiological features, which cannot always have the same radiological appearance in all CBCT radiographs. The presence of US can be a potential limitation for some surgical procedures in the maxillary sinus area. Each clinician should be aware of Underwood septa and their anatomical variations.

## Figures and Tables

**Figure 1 jpm-13-01258-f001:**
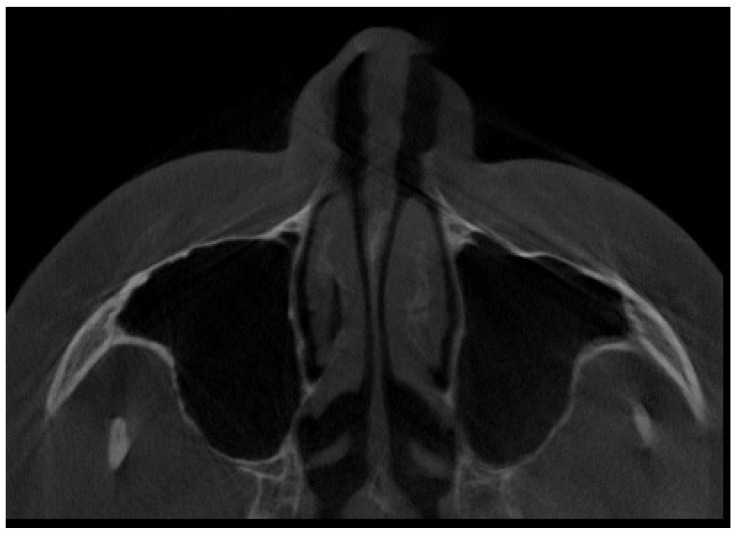
A CBCT axial view on healthy maxillary sinuses without any septal, bone, or other anatomical disturbances.

**Figure 2 jpm-13-01258-f002:**
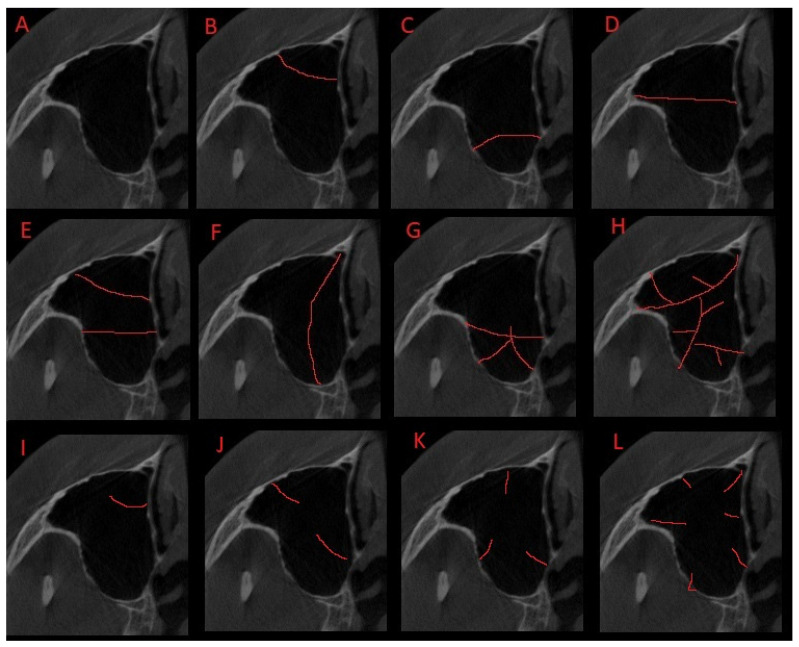
Authors’ proposal on maxillary sinuses with Underwood bone septa classification based on CBCT 20 × 20 axial scans. (**A**) Maxillary sinus without any bone/septal disturbances; (**B**) one septa in the anterior part of the sinus; (**C**) one septa in the posterior part of the sinus; (**D**) a single septa in the middle of the sinus; (**E**) two septa dividing the sinus into three components; (**F**) a vertical septa that divides the sinus into right and left, equal or not, parts; (**G**) located in part of the sinus, either anterior/posterior/middle part with some septa mimicking a clover leaf appearance; (**H**) maxillary sinus divided with multiple septa in a chaotic matter, like the spider-web appearance; (**I**) a single not full septa but a half septa in the maxillary sinus; (**J**) two half septa; (**K**) three half septa; (**L**) four or more half septa placed randomly in the maxillary sinuses. Red line – possible shape and location of a full or partial bony septa within the maxilalry sinus.

**Figure 3 jpm-13-01258-f003:**
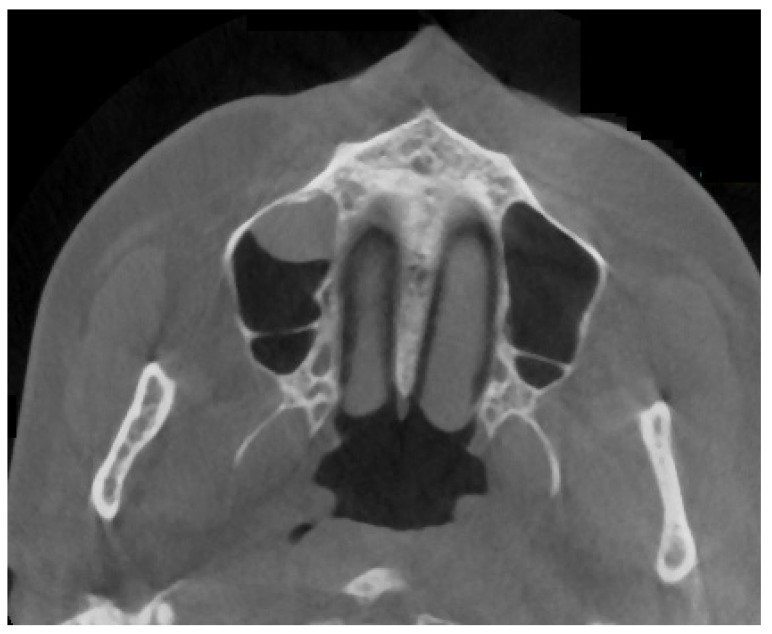
A posteriorly placed Underwood septa. Type C in the authors’ proposal.

**Figure 4 jpm-13-01258-f004:**
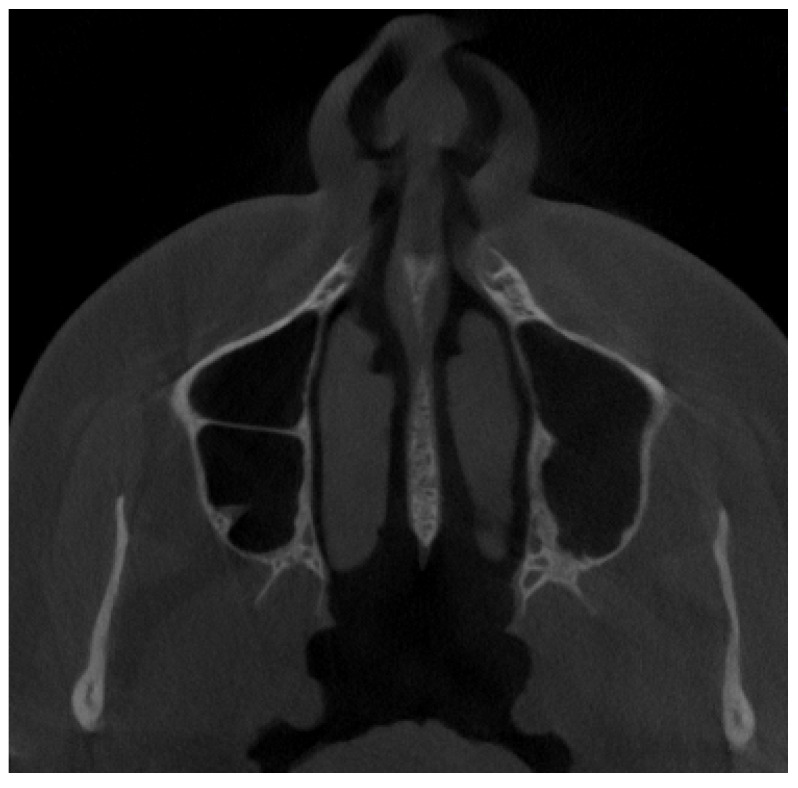
A single midline septa. Type D in the authors’ proposal.

**Figure 5 jpm-13-01258-f005:**
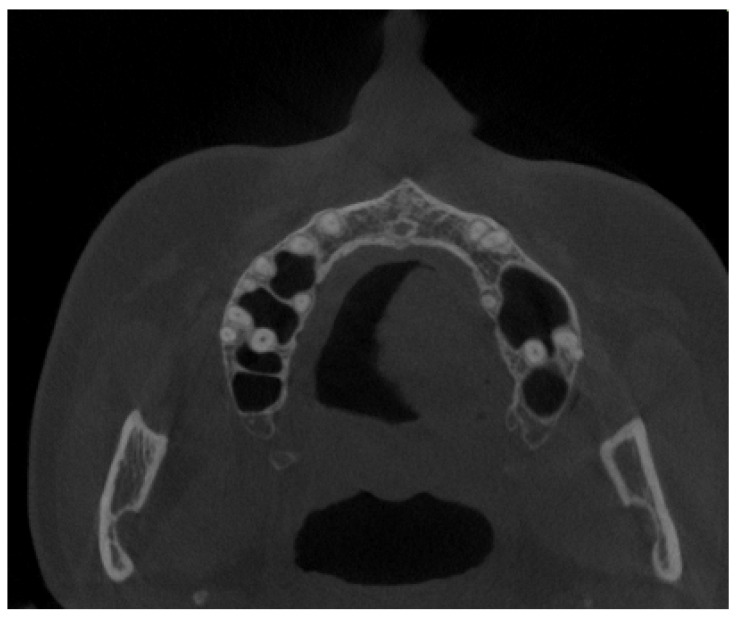
Anterior and posterior septa. Type E in the authors’ proposal—two septa dividing the sinus into three components.

**Figure 6 jpm-13-01258-f006:**
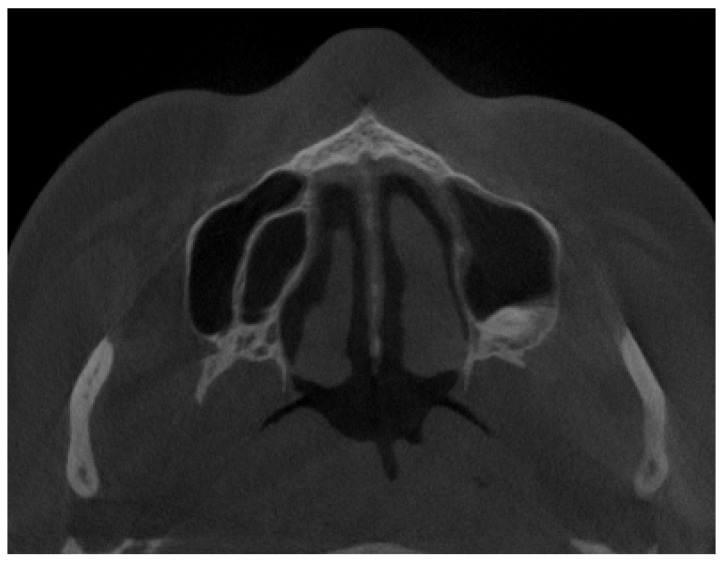
A vertical septa, presented in the authors’ proposal as a Type F vertical septa that divides the sinus into right and left, equal or not, parts.

**Figure 7 jpm-13-01258-f007:**
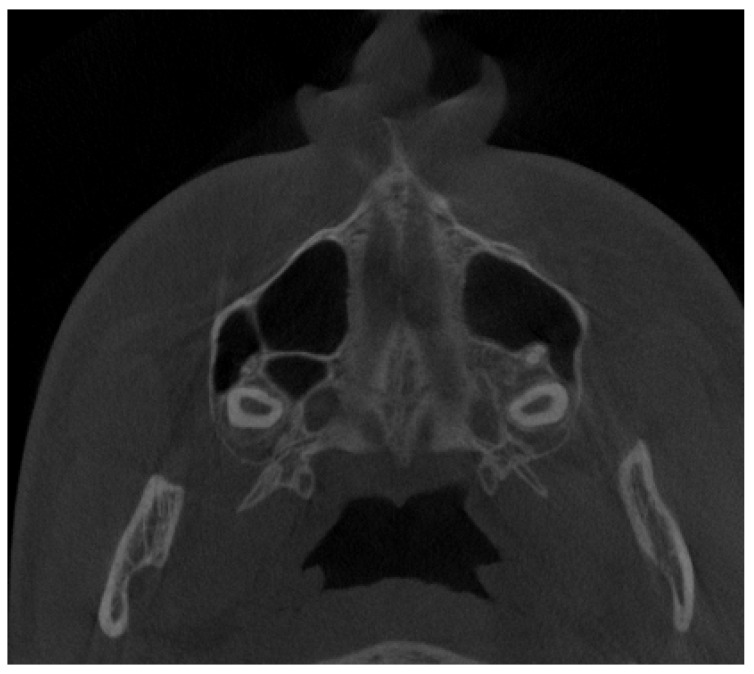
A clover leaf appearance, suggested by authors as Type G.

**Figure 8 jpm-13-01258-f008:**
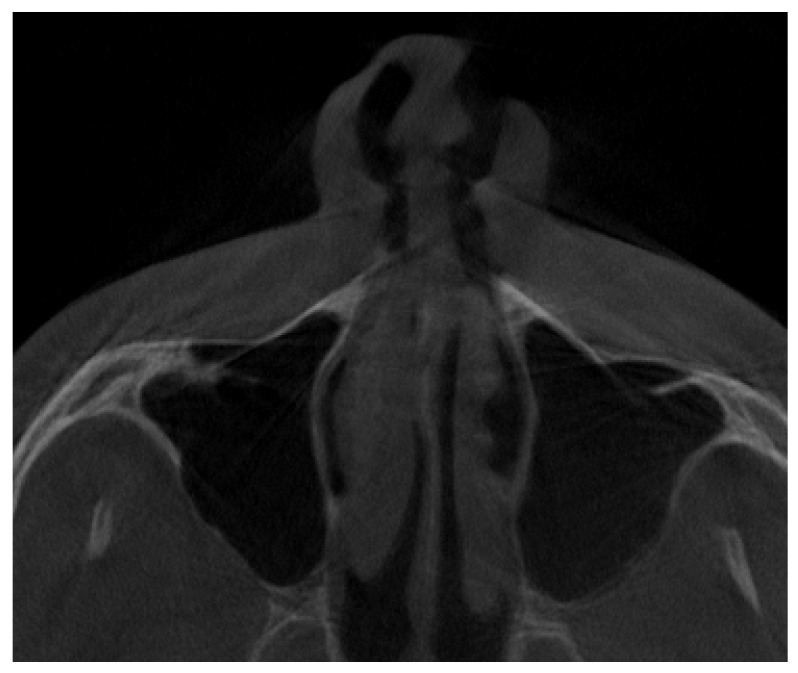
Half septa, visible in the left maxillary sinus, suggested by the authors as being similar to types I–L.

**Figure 9 jpm-13-01258-f009:**
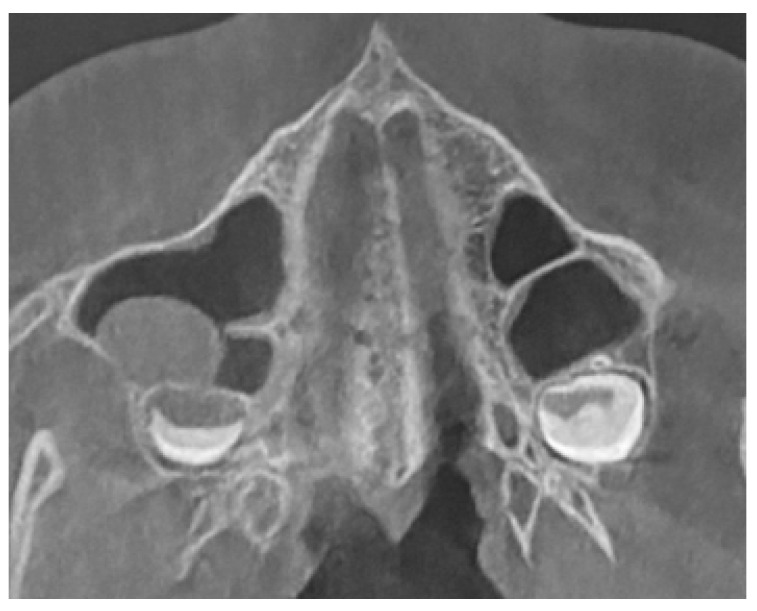
A different variant with complete septa in the left maxillary sinus and an incomplete septa in the right maxillary sinus coexisting with a retention cyst.

**Figure 10 jpm-13-01258-f010:**
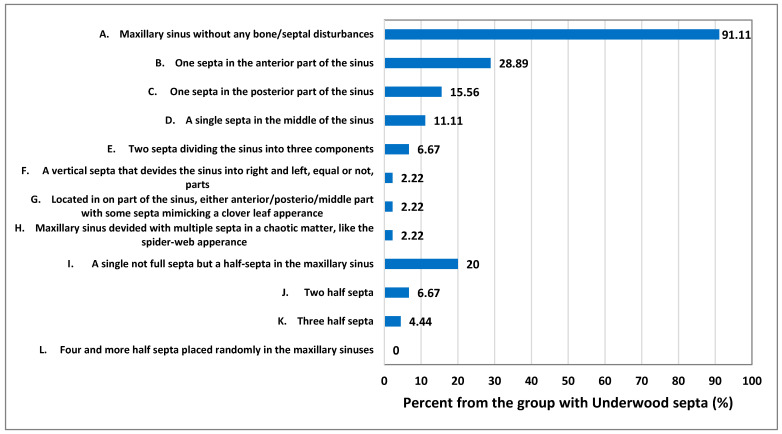
Distribution of findings of Underwood septa location and morphology (*n* = 45).

**Table 1 jpm-13-01258-t001:** Proposal of classification of maxillary sinuses Underwood bone septa based on CBCT 20 × 20 axial scans (*n* = 45).

Types of Classification	Number of Observations (Percent)
A. Maxillary sinus without any bone/septal disturbances	41 (91.11)
B. One septa in the anterior part of the sinus	13 (28.89)
C. One septa in the posterior part of the sinus	7 (15.56)
D. A single septa in the middle of the sinus	5 (11.11)
E. Two septa dividing the sinus into three components	3 (6.67)
F. A vertical septa that divides the sinus into right and left, equal or not, parts	1 (2.22)
G. Located in on part of the sinus, either anterior/posterior/middle part with some septa mimicking a clover leaf appearance	1 (2.22)
H. Maxillary sinus divided with multiple septa in a chaotic matter, like the spider-web appearance	1 (2.22)
I. A single not full septa but a half septa in the maxillary sinus	9 (20.00)
J. Two half septa	3 (6.67)
K. Three half septa	2 (4.44)
L. Four or more half septa placed randomly in the maxillary sinuses	0 (0.00)

**Table 2 jpm-13-01258-t002:** Relationship between demographical, clinical, or CT/CBCT radiological features and Underwood septa status (presence/absence) in the study group (*n* = 120).

Variable	Total Group (*n* = 120)	Patients without Underwood Septa (*n* = 75)	Patients with Underwood Septa (*n* = 45)	*p*-Value
Gender:				0.530
Male	47 (39.17)	31 (41.33)	16 (35.56)	
Female	73 (60.83)	44 (58.67)	29 (64.44)	
Age (years old)	34.06 ± 11.88	34.13 ± 12.31	33.95 ± 11.27	0.937
Dental status:				0.232
Full dentition	103 (85.83)	66 (88.00)	37 (82.22)	
Partial dentition	15 (12.50)	7 (9.33)	8 (17.78)	
Full edentulism	0 (0.00)	0 (0.00)	0 (0.00)	
Partial edentulism	2 (1.67)	2 (2.67)	0 (0.00)	
Healthy pneumatized sinus:				0.117
No	25 (20.83)	19 (25.33)	6 (13.33)	
Yes	95 (79.17)	56 (74.67)	39 (86.67)	
Silent sinus syndrome (SSS)				0.174
No	117 (97.50)	72 (96.00)	45 (100.00)	
Yes	3 (2.50)	3 (4.00)	0 (0.00)	
Mucosal thickening:				0.325
>5 mm	19 (15.83)	14 (18.67)	4 (8.89)	
<5 mm	23 (19.17)	13 (17.33)	10 (22.22)	
Without thickening	78 (65.00)	48 (64.00)	31 (68.89)	
Polypoidal mucosal thickening:				0.625
No	119 (99.17)	74 (98.67)	45 (100.00)	
Yes	1 (0.83)	1 (1.33)	0 (0.00)	
Retention cyst:				0.272
No	101 (84.17)	61 (81.33)	40 (88.89)	
Yes	19 (15.83)	14 (18.67)	5 (11.11)	
Sinus opacification:				0.203
Partial sinus opacification	9 (7.50)	8 (10.67)	1 (11.11)	
Complete sinus opacification	6 (5.00)	3 (4.00)	3 (6.67)	
Without opacification	85 (87.50)	64 (85.33)	41 (91.11)	
CRS presence:				0.599
No	116 (96.67)	72 (96.00)	44 (97.78)	
Yes	4 (3.33)	3 (4.00)	1 (2.22)	

## Data Availability

Availability of supporting data—the datasets used and/or analyzed during the current study are available from the corresponding author on reasonable request.

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
