# Peer review of "The CBCT Retrospective Study on Underwood Septa and Their Related Factors in Maxillary Sinuses—A Proposal of Classification"

_jpm, 2023, doi:10.3390/jpm13081258_

Round 1

Reviewer 1 Report

Dear Author, 

Dear Author,

Greetings

1.      Kindly do mention / define the maxillary septa or underwood septa prior to further details in the introduction

2.      Kindly mention the prevalence too in the beginning of the introduction

3.      It is mentioned that- CBCT is very important. In some cases, the scope of bone septa and their proximity to dental roots they can mimic a cyst when are not closely evaluated. Any surgery related to… kindly revise

4.      It is mentioned that- the blood supply to the maxillary sinus. Some studies indicate decreased blood supply in elderly, edentulous patients [12-20]… kindly comment whether the former is an observation post surgery or otherwise.. kindly revise as it is not clear

5.      Dear authors, kindly verify the flow of the narration in the introduction . kindly verify whether rearrangement is required as different patterns and classifications are mentioned earlier on in the introduction rather towards the end directing it towards the aim of the study. kindly verify

6.      It is mentioned that- The following research is a retrospective study on CBCT radiographs from Authors Own database … kindly comment whether it is an institutional/ clinical practice database. Kindly be specific

7.      Kindly comment the time period – the duration from when to when were the records considered as the study is retrospective in nature

8.      Methodology needs to be narrated in past tense. kindly revise

9.      It is mentioned that- A total of 120 CBCT scans were evaluated.. kindly note that the former comprises of results, hence kindly exclude from methodology

10.   It is mentioned that- thickness between 0,070mm to 0,3 mm… kindly verify whether it is decimal/ comma

11.  It is mentioned that- sinus septal and bone disturbance only in axial planes of the CBCT to ensure that the presented Authors' proposal will be a good, simplified, and fast diagnostics method for this bone pathology…

a.       Kindly comment whether it is considered as a bone pathology

b.       Kindly revise the narration too

12.  Kindly comment whether the presence of these septa is seen in all as the selection criteria states that maxillary sinuses were studied, but did not confirm the presence of septae in all

13.  Dear authors, kindly verify the methodology narration wrt the drafting of the classification proposal as the narration concentrates and emphasize wrt the septae while its presence is not mandatory as stated below. Kindly revise and narrate accordingly

a.       The authors described each maxillary sinus septal and bone disturbance only in axial planes of the CBCT to ensure that the presented Authors' proposal will be a good,simplified, and fast diagnostics method for this bone pathology. The

b.       The following classification proposal is shown: - A-maxillary sinus without any bone/septal disturbances

14.  It is mentioned that- On 120 CBCT radiographs evaluation, a total of 37,5% were only related to both full.. kindly verify the values

15.  Classification proposal-  dear authors, kindly exclude the beginning of the narration as the introduction and the methodology have addressed the same. Kindly narrate wrt the results obtained alone and in specific manner

16.  Clinical and surgical cosiderations- kindly comment on the relevance of the presented narration in the results. kindly note that the narration presented can be addressed in the discussion . instead, kindly comment on the no: of cases which might need an intervention and the importance of the anatomical consideration wrt the sepate in regards to the former instead. Kindly be specific based on the data accumulated and derive the findings accordingly

17.  Repetitions are observed in the manuscript narration. Kindly verify and revise

18.  Kindly revise the figure legends

19.  Dear authors, kindly exclude repetitive stating the phrase - the authors in the narration as methodology defines the database considered and the aim for classification proposal by the authors.. kindly revise

Regards

No major concerns noted.

Author Response

Dear reviewer, thank you for valuable comments and insight. We greately appriciate your knowledge and hints to make this paper even better original paper. 

Q1
1.      Kindly do mention / define the maxillary septa or underwood septa prior to further details in the introduction
R1
Dear reviewer, thank you for a great comments. Added ".... The Underwood septa (US) was first described by Underwood in his anatomical study on maxillary sinuses (MS) [1]. The US is a part of a thin bone wall-shaped structure inside the maxillary sinus, with various differences in shape, size, its position and bone volume. The differences in MS  (....)"

Q2
2.      Kindly mention the prevalence too in the beginning of the introduction
R2
Dear reviewer, thank you for a great comments. Added ().....Its prevalence varies from 12-32% or more, depending on the studies [1-8]. (...)

Q3
3.      It is mentioned that- CBCT is very important. In some cases, the scope of bone septa and their proximity to dental roots they can mimic a cyst when are not closely evaluated. Any surgery related to… kindly revise
R3
Dear reviewer, thank you for a great comments. Added and changed. (...)"The importance of their identification in CBCT is very important, especially if radiological evaluation in related with future surgeries. (...)"

Q4
4.      It is mentioned that- the blood supply to the maxillary sinus. Some studies indicate decreased blood supply in elderly, edentulous patients [12-20]… kindly comment whether the former is an observation post surgery or otherwise.. kindly revise as it is not clear
R4
Dear reviewer, thank you for a great comments. Added and changed - deleted and re-written

Q5
5.      Dear authors, kindly verify the flow of the narration in the introduction . kindly verify whether rearrangement is required as different patterns and classifications are mentioned earlier on in the introduction rather towards the end directing it towards the aim of the study. kindly verify
R5
Dear reviewer, thank you for a great comments. It has been verified. 

Q6
6.      It is mentioned that- The following research is a retrospective study on CBCT radiographs from Authors Own database … kindly comment whether it is an institutional/ clinical practice database. Kindly be specific
R6
Dear reviewer, thank you for a great comments. It has been verified. 
The following research is a retrospective study on CBCT radiographs from Authors Own clinical practice database. 

Q7
7.      Kindly comment the time period – the duration from when to when were the records considered as the study is retrospective in nature
R7
Dear reviewer, thank you for a great comments. It has been verified. 

Q8
8.      Methodology needs to be narrated in past tense. kindly revise
R8
Dear reviewer, thank you for a great comments. It has been verified. 

Q9
9.      It is mentioned that- A total of 120 CBCT scans were evaluated.. kindly note that the former comprises of results, hence kindly exclude from methodology
R9
Dear reviewer, thank you for a great comments. It has been removed 

Q10
10.   It is mentioned that- thickness between 0,070mm to 0,3 mm… kindly verify whether it is decimal/ comma
R10
Dear reviewer, thank you for a great comments. It has been changed "."

Q11
11.  It is mentioned that- sinus septal and bone disturbance only in axial planes of the CBCT to ensure that the presented Authors' proposal will be a good, simplified, and fast diagnostics method for this bone pathology…
a.       Kindly comment whether it is considered as a bone pathology
b.       Kindly revise the narration too
R11
Dear reviewer, thank you for a great comments. It has been changed. Pathology was deleted. (...)"and fast diagnostics method for this bone anatomical variations "(...)

Q12
12.  Kindly comment whether the presence of these septa is seen in all as the selection criteria states that maxillary sinuses were studied, but did not confirm the presence of septae in all
R11
Dear reviewer, thank you for a great comments. Table 2 describes the sinus without any walls/hemi-walls in the maxillary sinuses

Q13
13.  Dear authors, kindly verify the methodology narration wrt the drafting of the classification proposal as the narration concentrates and emphasize wrt the septae while its presence is not mandatory as stated below. Kindly revise and narrate accordingly
a.       The authors described each maxillary sinus septal and bone disturbance only in axial planes of the CBCT to ensure that the presented Authors' proposal will be a good,simplified, and fast diagnostics method for this bone pathology. The
b.       The following classification proposal is shown: - A-maxillary sinus without any bone/septal disturbances
R13
Dear reviewer, thank you for a great comments. It has been changed - SEPTA A - is a healthy sinus without any septa, half septa or any other related pathology

Q14
14.  It is mentioned that- On 120 CBCT radiographs evaluation, a total of 37,5% were only related to both full.. kindly verify the values
R14
Dear reviewer, thank you for a great comments.
On 120 CBCT radiographs evaluation, a total of 37,5% were only related to both full (n=31) and/or partial bony septa (n=14) in the maxillary sinus cavity (p>0.05) (Tab.2.). 

Q15
15.  Classification proposal-  dear authors, kindly exclude the beginning of the narration as the introduction and the methodology have addressed the same. Kindly narrate wrt the results obtained alone and in specific manner
R15
Dear reviewer, thank you for a great comments. It has been changed 

Q16
16.  Clinical and surgical cosiderations- kindly comment on the relevance of the presented narration in the results. kindly note that the narration presented can be addressed in the discussion . instead, kindly comment on the no: of cases which might need an intervention and the importance of the anatomical consideration wrt the sepate in regards to the former instead. Kindly be specific based on the data accumulated and derive the findings accordingly
R16
Dear reviewer, thank you for a great comments. It has been changed 
The presence of full or partial Underwood bone septa in maxillary sinuses might in-fluence the result of some surgeries within the sinus itself. A half septa occurrence might lead to a bigger possibility of a torn in the lifted mucous membrane of the sinus, while a full septa can be a significant anatomical obstacle to perform a good lift procedure. 

Q17
17.  Repetitions are observed in the manuscript narration. Kindly verify and revise
R17
Dear reviewer, thank you for a great comments. It has been changed 

Q18
18.  Kindly revise the figure legends
R18
Dear reviewer, thank you for a great comments. It has been changed 

Q19
19.  Dear authors, kindly exclude repetitive stating the phrase - the authors in the narration as methodology defines the database considered and the aim for classification proposal by the authors.. kindly revise
R19
Dear reviewer, thank you for a great comments. It has been changed 

Reviewer 2 Report

The manuscript addresses a modern and multidisciplinary research topic with impact on diagnosis of any pathology and surgical procedures of maxillary sinus. The authors investigated the patterns of maxillary sinus bone septa and present their own proposal for this anatomical variations classification based on CBCT axial scans. Also, the authors proposed to correlate the co-existing maxillary sinus features with the so-called Underwood septa (US). The originality of the manuscript is represented by the study conclusions. The authors mentioned that “the occurrence of US is not statistically related to any clinical, radiological, or pathological condition within the sinus (p>0,05). Furthermore, more full or partial appearance of US was found in female patients.” The researchers recommended to any clinician to be aware of the US and their anatomical variations because the presence of US can be a potential limitation for some surgical procedures in the maxillary sinus area. Also, the researchers pointed out the limitations of the study because the research includes a small number of CBCT database and patients with various scopes of diseases and syndromes within the facial skeleton diagnosed in CBCT.

Strengths of manuscript

The manuscript corresponds to the stated purpose and objectives of the journal.

The title accurately reflects the content of the paper.

The abstract is structured as an accurate synopsis of the paper.

The introduction presents in detail the current state of knowledge on the approached subject and highlights why this research is important. It is define the purpose of the work and its significance. Also, the introduction includes 22 relevant references, out of which 11 are published after 2018.

The manuscript contains a complex, but correctly designed and technically sound research method. The authors' proposal on classification of US bone of maxillary sinuses based on CBCT is sustained by 8 figures. The research steps are clearly presented in the “Material and Methods” chapter. The instruments, research equipment and software programs are described in sufficient details to allow another researcher to reproduce the results.

The performed analyses are appropriate. The research results are presented at higher standards, including 2 tables and 1 figure. The images are significant and suggestive for the research subject. The results of the research are concisely and systematically present. All figures are elaborated according to authors’ guideline.

The discussions present an interpretation of the results in perspective of previous studies and of the aim of the study. The comparison of the research results was made taking into account another 17 references, out of which 6 are published after 2017. The findings and their implications are discussed in the broadest context possible.  

The conclusions are presented in generally manner and highlight the research implications on surgical procedures in the maxillary sinus area. They are interesting for the readership of the journal.

The references are in accordance with the studied topic. The manuscript contains 39 references, most representing studies published after the year 2000.

Weakness of the manuscript

-The abstract is too longer. It has 213 words but the authors’ guideline mentions that the abstract should not be longer than 200 words.

-It would be better if, chapter “2.2. Patients’ characteristics” should be called CBCT characteristics.

-In the chapter “Results” point “3.1. Relationship between demographic, clinical or CT/CBCT radiological features and Underwood septa status” is mentioned the figure 9 in the phrase “The following suggests that the number of teeth present in the dental arch and the patient's age doesn't correlate with the septal occurrence along with additional symptoms (Tab. 1;2)(Fig.9)”.

Also, figure 9 is mentioned before chapter “3.3. Clinical and surgical considerations”, as follows: “Figure 9. Distribution of findings of Underwood septa location and morphology (n=45)”.

The text below two images has the name of figure 9, as follows:

Figure 9. A different variant with complete septa in the left maxillary sinus and an incomplete septa in the right maxillary sinus coexisting with an retention cyst.

Figure 9. Distribution of findings of Underwood septa location and morphology (n=45).

Author Response

Dear reviewer, thank you for valuable comments and insight. We greately appriciate your knowledge and hints to make this paper even better original paper. 
Q1
The manuscript addresses a modern and multidisciplinary research topic with impact on diagnosis of any pathology and surgical procedures of maxillary sinus. The authors investigated the patterns of maxillary sinus bone septa and present their own proposal for this anatomical variations classification based on CBCT axial scans. Also, the authors proposed to correlate the co-existing maxillary sinus features with the so-called Underwood septa (US). The originality of the manuscript is represented by the study conclusions. The authors mentioned that “the occurrence of US is not statistically related to any clinical, radiological, or pathological condition within the sinus (p>0,05). Furthermore, more full or partial appearance of US was found in female patients.” The researchers recommended to any clinician to be aware of the US and their anatomical variations because the presence of US can be a potential limitation for some surgical procedures in the maxillary sinus area. Also, the researchers pointed out the limitations of the study because the research includes a small number of CBCT database and patients with various scopes of diseases and syndromes within the facial skeleton diagnosed in CBCT.
R1
Dear reviewer, thank you for a great comments. The occurence of Underwood Septa are indeed some sirious anatomical consition that shoule be remebered. Each surgeon or laryngologisy should be aware to identify this sinus wall anomaly and adress its possible outcomes.

Q2
Strengths of manuscript
The manuscript corresponds to the stated purpose and objectives of the journal.
The title accurately reflects the content of the paper.
The abstract is structured as an accurate synopsis of the paper.
R2
Dear reviewer, thank you for a great comments.

Q3
The introduction presents in detail the current state of knowledge on the approached subject and highlights why this research is important. It is define the purpose of the work and its significance. Also, the introduction includes 22 relevant references, out of which 11 are published after 2018.
The manuscript contains a complex, but correctly designed and technically sound research method. The authors' proposal on classification of US bone of maxillary sinuses based on CBCT is sustained by 8 figures. The research steps are clearly presented in the “Material and Methods” chapter. The instruments, research equipment and software programs are described in sufficient details to allow another researcher to reproduce the results.
R3
Dear reviewer, thank you for a great comments.

Q4
The performed analyses are appropriate. The research results are presented at higher standards, including 2 tables and 1 figure. The images are significant and suggestive for the research subject. The results of the research are concisely and systematically present. All figures are elaborated according to authors’ guideline.
R4
Dear reviewer, thank you for a great comments.

Q5
The discussions present an interpretation of the results in perspective of previous studies and of the aim of the study. The comparison of the research results was made taking into account another 17 references, out of which 6 are published after 2017. The findings and their implications are discussed in the broadest context possible.  
R5
Dear reviewer, thank you for a great comments.

Q6
The conclusions are presented in generally manner and highlight the research implications on surgical procedures in the maxillary sinus area. They are interesting for the readership of the journal.
The references are in accordance with the studied topic. The manuscript contains 39 references, most representing studies published after the year 2000.
R6
Dear reviewer, thank you for a great comments.

Q7
Weakness of the manuscript
-The abstract is too longer. It has 213 words but the authors’ guideline mentions that the abstract should not be longer than 200 words.
R7
Dear reviewer, thank you for a great comments. The abstract was re-arranged and shortened. Its 199 words

Q8
-It would be better if, chapter “2.2. Patients’ characteristics” should be called CBCT characteristics.
R8
Dear reviewer, thank you for a great comments. Chapter name CHanged.

Q9
-In the chapter “Results” point “3.1. Relationship between demographic, clinical or CT/CBCT radiological features and Underwood septa status” is mentioned the figure 9 in the phrase “The following suggests that the number of teeth present in the dental arch and the patient's age doesn't correlate with the septal occurrence along with additional symptoms (Tab. 1;2)(Fig.9)”.
R9
Dear reviewer, thank you for a great comments. Its changed now.

Q10
Also, figure 9 is mentioned before chapter “3.3. Clinical and surgical considerations”, as follows: “Figure 9. Distribution of findings of Underwood septa location and morphology (n=45)”.
R10
Dear reviewer, thank you for a great comments.Its changed now.

Q11
The text below two images has the name of figure 9, as follows:
Figure 9. A different variant with complete septa in the left maxillary sinus and an incomplete septa in the right maxillary sinus coexisting with an retention cyst.
Figure 9. Distribution of findings of Underwood septa location and morphology (n=45).
R11
Dear reviewer, thank you for a great comments.Its changed now.